# A New Framework for Modelling Fine Sediment Transport in Rivers Includes Flocculation to Inform Reservoir Management in Wildfire Impacted Watersheds

**Micheal Stone [1], Bommanna G. Krishnappan [2],\*, Uldis Silins [3], Monica B. Emelko [4], Chris H. S. Williams [3], Adrian L. Collins [5] and Sheena A. Spencer [6]**

1   Environmental Studies, University of Waterloo, Waterloo, ON N2L 3G1, Canada; mstone@uwaterloo.ca
2   Environment Canada, Burlington, ON L7R 4A6, Canada
3   Department of Renewable Resources, University of Alberta, Edmonton, AB T6G 2G7, Canada; usilins@ualberta.ca (U.S.); chw1@ualberta.ca (C.H.S.W.)
4   Department of Civil and Environmental Engineering, University of Waterloo, Waterloo, ON N2L 3G1, Canada; mbemelko@uwaterloo.ca
5   Sustainable Agriculture Sciences Department, Rothamsted Research, North Wyke, Okehampton EX20 2SB, UK; Adrian.collins@rothamsted.ac.uk
6   Ministry of Forests, Lands, Natural Resource Operations and Rural Development, Government of British Columbia, Penticton, BC V2A 7C8, Canada; sheena.spencer@gov.bc.ca
\*   Correspondence: krishnappan@sympatico.ca

**Abstract:** Fine-grained cohesive sediment is the primary vector for nutrient and contaminant redistribution through aquatic systems and is a critical indicator of land disturbance. A critical limitation of most existing sediment transport models is that they assume that the transport characteristics of fine sediment can be described using the same approaches that are used for coarse-grained non-cohesive sediment, thereby ignoring the tendency of fine sediment to flocculate. Here, a modelling framework to simulate flow and fine sediment transport in the Crowsnest River, the Castle River, the Oldman River and the Oldman Reservoir after the 2003 Lost Creek wildfire in Alberta, Canada was developed and validated. It is the first to include explicit description of fine sediment deposition/erosion processes as a function of bed shear stress and the flocculation process. This framework integrates four existing numerical models: MOBED, RIVFLOC, RMA2 and RMA4 using river geometry, flow, fine suspended sediment characteristics and bathymetry data. Sediment concentration and particle size distributions computed by RIVFLOC were used as the upstream boundary condition for the reservoir dispersion model RMA4. The predicted particle size distributions and mass of fine river sediment deposited within various sections of the reservoir indicate that most of the fine sediment generated by the upstream disturbance deposits in the reservoir. Deposition patterns of sediment from wildfire-impacted landscapes were different than those from unburned landscapes because of differences in settling behaviour. These differences may lead to zones of relatively increased internal loading of phosphorus to reservoir water columns, thereby increasing the potential for algae proliferation. In light of the growing threats to water resources globally from wildfire, the generic framework described herein can be used to model propagation of fine river sediment and associated nutrients or contaminants to reservoirs under different flow conditions and land use scenarios. The framework is thereby a valuable tool to support decision making for water resources management and catchment planning.

**Keywords:** cohesive sediment; erosion; water supply; turbidity; gravel bed river; ingress; watershed management; source water protection; climate change adaptation; landscape disturbance

## 1. Introduction

Forested regions provide approximately 86% of surface water supplies in the United States (Caldwell et al. [1]) and more than 58% for the largest Canadian urban and ru-

ral communities as well as the majority of Canadian Indigenous communities. Rivers draining forested landscapes are important for the provision of high-quality source water and support of healthy aquatic ecosystems. While various anthropogenic disturbances (e.g., harvesting, recreational use, land clearing for agriculture or resource extraction) in these critical water-bearing landscapes can alter erosion and runoff to, and subsequent sedimentation within, receiving waters (Kastridis and Kamperidou [2], Vacca et al. [3]), the increasing frequency and severity of landscape disturbance by wildfire has raised urgent concerns about degraded and more variable water quality and its implications for the provision of safe drinking water (Vörösmarty [4]; Emelko et al. [5]; International Panel on Climate Change (IPCC) [6]). Accordingly, watershed management is directly linked to national security in some regions (Caldwell et al. [1]). National and international commitment to source water protection in forested watersheds has been increasingly advocated (Vörösmarty et al. [7]; Emelko and Sham [8]).

Wildfire is the most severe large-scale landscape disturbance in critical forested source water regions (Emelko and Sham [8]; Vose et al. [9]; Khan et al. [10]). Recent increases in the size and severity of wildfires related to climate warming (Westerling et al. [11]; Flannigan et al. [12]) have been shown to degrade terrestrial ecosystems, ecological processes and functions, and surface water quality (Benda et al. [13]; Khan et al. [10]), whilst threatening human life and property (Kinoshita et al. [14]).

Wildfire impacts on water quality vary because of differences in physiographic setting as well as hydro-climatic and landscape factors such as wind speed, moisture conditions, and vegetation type (Bisson et al. [15]; Vörösmarty et al. [7]; Silins et al. [16]; Lucas-Borja et al. [17]; Plaza-Alvarez et al. [18]). The effects of severe wildfire on water quality have been observed at the watershed scale (Emmerton et al. [19]; Silins et al. [20]; Emelko et al. [21]) wherein increases in sediment yields and instream sedimentation have been measured (Benda et al. [13]; Tobergte and Curtis [22]). Wildfire can exacerbate the impact of extreme rain events, which can mobilise and transport significant amounts of sediment (López-Vicente et al. [23], Malmon et al. [24]). Critically, in regions underlain by glacial deposits, such as the eastern slopes of the Rocky mountains in Alberta, Canada, fine sediment delivery from terrestrial to aquatic systems can be elevated and prolonged (Silins et al. [25]; Stone et al. [26]); it also contributes to the downstream transfer and fate of sediment-associated phosphorus which, in turn, influences reservoir water quality (Stone et al. [26]; Silins et al. [20]; Emelko et al. [21]).

Sediment is the primary vector for nutrient and contaminant redistribution through aquatic systems (Horowitz and Elrick [27]; Ongley et al. [28]; Chapman et al. [29]) and is a critical indicator of land disturbance (Walling and Collins [30]). Excessive amounts of fine sediment can reduce light transmission in high quality streams, decrease flow through interstitial gravels and lower oxygen supply in spawning habitat (Collins et al. [31]; Wood and Armitage [32]). Accurate representation of fine sediment transport informs the fate and bioaccumulation of many toxic substances and the availability of limiting nutrients such as phosphorus, which contribute to eutrophication in aquatic systems. Accordingly, there is an important need to model fine sediment transport in aquatic systems as robustly as possible.

A critical limitation of most existing sediment transport models is that they assume that the transport characteristics of fine-grained cohesive sediment can be described using the same approaches that are used for coarse-grained non-cohesive sediment, thereby ignoring the fundamental tendency of fine sediment to flocculate (Partheniades [33]; Krone [34]; Mehta [35]; Lick [36]). Flocculation strongly influences the transport properties (porosity, density, settling velocity) and fate of fine sediment and associated contaminants (Lau and Krishnappan [37], Krishnappan [38,39]). Thus, while coarse-grained sediments undergo simultaneous erosion and deposition during transport at constant bed shear stress in aquatic systems, the simultaneous erosion and deposition of fine sediments is not possible and they undergo either deposition or erosion at certain bed shear stresses, but not both (Partheniades and Kennedy [40]; Mehta and Partheniades [41]; Lau and Krishnappan [37];

Krishnappan [38]). If simultaneous erosion and deposition are assumed in transport modelling, fine sediment and associated contaminant concentrations will be under-predicted; in contrast, appropriate representation of mutually exclusive erosion and deposition will preserve the comparatively high concentrations of sediment and associated contaminants that will transport over relatively longer distances from the source (Krishnappan [38,39]). Improved understanding and representation of the mobilisation, flocculation and transport dynamics of fine sediment in watersheds is therefore an essential pre-requisite for land managers seeking to evaluate the fate and impacts of diffuse source pollution resulting from both anthropogenic and natural landscape disturbances (Walling and Collins [30]; Emelko et al. [5]).

To date, modelling efforts to describe disturbance impacts on water have been largely limited to simulating hydrologic and erosion responses (e.g., streamflow, infiltration, runoff). For example, the Soil and Water Assessment (SWAT) model has been applied to compute change in streamflows after wildfire (Rodrigues et al. [42]). Several physically-based transport models have been developed to describe coarse-grained sediment erosion in wildfire impacted landscapes at the plot and hillslope scales (Moody et al. [43]) and landscape evolution models have been applied to simulate gullying, landslides and debris flows in small watersheds (Lancaster et al. [44]; Istanbulluoglu et al. [45]). Frameworks have also been developed to simulate sediment delivery processes from runoff-generated debris flows to reservoirs at the watershed scale; however, they are not data driven and only involve empirical calculation of debris flow volumes and expert-based assumptions regarding fine sediment delivery (Laghans et al. [46]). Post-fire water quality has also been simulated using the MIKE Hydro Basin water quality model with ECOLab (Santos et al. [47]). The MIKE ECOLab module conducts simplified water quality simulations solving generic ordinary differential equations based on user inputs (rather than hard-coded calculations) (Danish Hydraulic Institute [48]). While the MIKE Hydro Basin module does not include either cohesive (i.e., fine-grained) or non-cohesive sediment, the MIKE Hydro River module includes a 1D computational platform for modelling sediment transport. Cohesive sediments are treated as suspended load—cohesive and non-cohesive sediments are transported in the same manner, but cohesive sediments use different erosion and deposition functions (Danish Hydraulic Institute [49]). Critically, this platform requires user definition of the fraction of total load that is suspended (i.e., fine-grained) and does not include flocculation (Danish Hydraulic Institute [49]). Thus, other investigations of climate and land use change impacts on sediment transport that have coupled hydrologic models like SWAT with the MIKE platform (Anand et al. [50]) also suffer from the same important limitation of disregarding the flocculation process.

From a management perspective, there is a critical need to develop and test process-based, as opposed to risk-based, models that simulate fine sediment transport dynamics in rivers and downstream receptors for a range of land disturbance types (Walling et al. [51]; Walling and Collins [30]; Daniel et al. [52]). This need arises from the capacity of process understanding to help inform targeted intervention above and beyond the spatial information generated by risk-based approaches. Given the significance of fine sediment for contaminant transport (Horowitz and Elrick [27]; Ongley et al. [28]) and the influence of flocculation (Lau and Krishnappan [37], Krishnappan [38,39]) on its redistribution and fate in aquatic systems, it is necessary to advance sediment transport models to incorporate flocculation processes explicitly (Summer and Walling [53]). At present, models that specifically quantify fine sediment transport processes and flocculation in river systems at large basin scales are scant. To address this research gap, the objectives of this work were to: (1) formulate a framework that includes explicit description of deposition/erosion processes as a function of bed shear stress and the flocculation process to model fine sediment transport in rivers to inform reservoir management in wildfire impacted watersheds, and; (2) demonstrate the utility of this framework to quantify sediment fluxes to reservoirs and inform post-fire reservoir management. The modelling framework reported herein is applied to three rivers in the Oldman watershed located immediately upstream of the Old-

man Reservoir. Detailed hydrometric and sediment monitoring surveys were conducted in the upper part of the watershed to calibrate flow and fine sediment transport models. Long term hydrometric and sediment data from reference (unburned) and wildfire impacted tributaries of the Crowsnest River are used to demonstrate the utility of the framework to quantify sediment transport and depositional fluxes in the river and reservoir.

## 2. Materials and Methods

### 2.1. Study Area

The study was conducted in the upper Oldman River watershed situated along the front range of the Rocky Mountains in southwest Alberta, Canada (Figure 1). The three main rivers that drain the watershed are the Oldman River from the north (area = 1923 km$^2$, mean elevation = 1741 m.a.s.l (3093–1110), mean basin slope = 44% (928–0%)), the Crowsnest River from the west (area = 1006 km$^2$, mean elevation = 1554 m.a.s.l (2803–1110), mean basin slope = 40% (611–0%)) and the Castle River from southwest (area = 1224 km$^2$, mean elevation = 1623 m.a.s.l (2737–1076), mean basin slope = 47% (891–0%)). All three rivers discharge into the Oldman Reservoir, which was created in 1993 when the Oldman Dam was built as an instream storage facility (~500 million m$^3$) for irrigation, power generation and recreational activities. The reservoir is located about 30 km downstream of the town of Blairmore in the Crowsnest Pass, where there was a severe forest fire (Lost Creek Wildfire) in 2003. The reservoir is ~20 km in length, with a maximum depth of ~65 m and a maximum width of ~2 km.

Mean annual precipitation and air temperature (town of Coleman, Alberta in the central study region) is 582 mm/yr and 3.6 °C, respectively with mean summer (July) and winter (December) air temperatures of 14.3 and −7.4 °C, respectively. Land cover in the study region spans high elevation alpine (exposed sedimentary bedrock and alpine shrubs), sub-alpine forests (dominated by *Abies lasiocarpa* and *Picea engelmannii*) and lower elevation montane forests (dominated by *Pinus contorta var. latifolia* and *Pseudotsuga menziesii var. glauca*) in western and central regions, with mixed native rangeland vegetation the lower elevation eastern region.

In 2003, the Lost Creek fire severely burned a near contiguous area of 21,065 ha on the eastern slopes of the Rocky Mountains in southern Alberta (Silins et al. [25]) which is one of the highest source-water yielding regions in the province (Figure 1). The subsequent post-fire hydrological changes dramatically increased sediment production (Silins et al. [25]) and accelerated the propagation of cohesive sediment and associated contaminants to downstream environments (Stone et al. [26]; Emelko et al. [21]) including the Oldman Reservoir. The new modelling framework described herein is applied to route fine-grained (<62.5 μm) sediments from the upper watershed to the reservoir to simulate the response to the wildfire disturbance.

### 2.2. Modelling Framework

The modelling framework shown schematically in Figure 2 incorporates four existing models: a river flow model (MOBED) that predicts the unsteady and non-uniform flows in rivers under mobile boundary conditions (Krishnappan [54–56]); a fine sediment transport and dispersion model (RIVFLOC) that calculates the dispersion and flocculation of fine-grained sediment in rivers (Krishnappan [57]); a reservoir flow model (RMA2) that simulates two dimensional flow fields (Donnell et al. [58]), and; a water quality model (RMA4) that can compute the dispersion and transport of fine sediment within a reservoir (Letter et al. [59]). The reservoir models RMA2 and RMA4 are part of the TABS-MD modelling system which is linked to an SMS user interface developed by AQUAVEO (www.aquaveo.com (accessed on 21 August 2021)). Specific components of the framework are briefly described below.

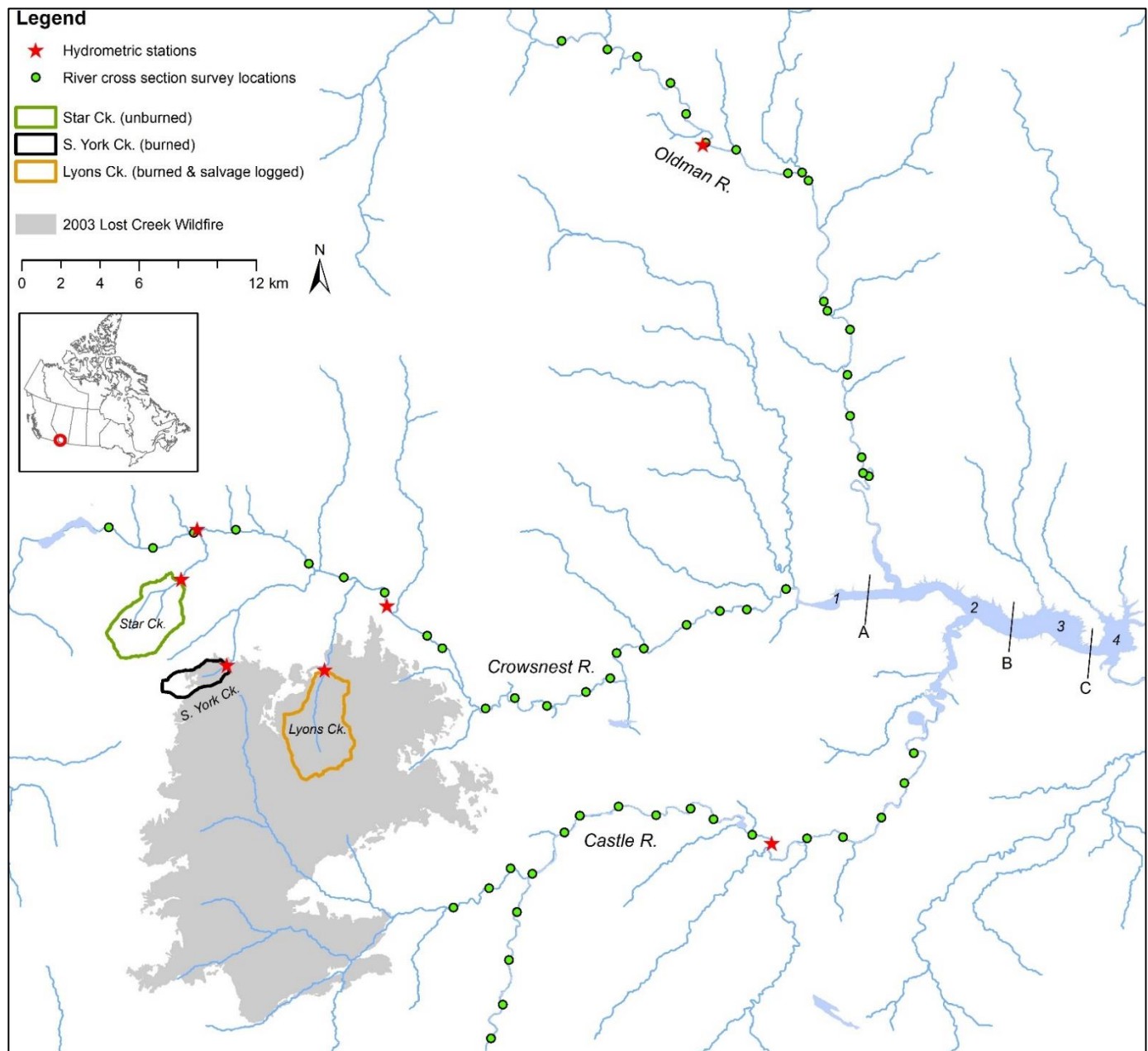

**Figure 1.** Study area. Upper Oldman River basin including the upper Oldman, Crowsnest and Castle Rivers, locations of cross section surveys, Water Survey of Canada and Southern Rockies Watershed Project hydrometric stations and three gauged unburned, burned, and post-fire salvage logged catchment tributaries to the main stem of the Crowsnest River. Inset shows study region in southwest Alberta, Canada.

2.2.1. Mobile Boundary Flow Model—MOBED

MOBED is an unsteady, mobile boundary flow model based on the numerical solution of St. Venant's equations and a sediment continuity equation (Krishnappan [54]). MOBED calculates flow rate, flow depth and bed shear stress as a function of time and distance in alluvial channels for a given upstream boundary flow condition. The model calculates the transport of coarse-grained bed sediment and resulting changes in the bed level for a full range of feasible flow conditions. MOBED uses a generalized friction factor equation that can be applied to various river types and bed forms (Krishnappan [54–56]). Specific details of the model can be found in Krishnappan [54–56].

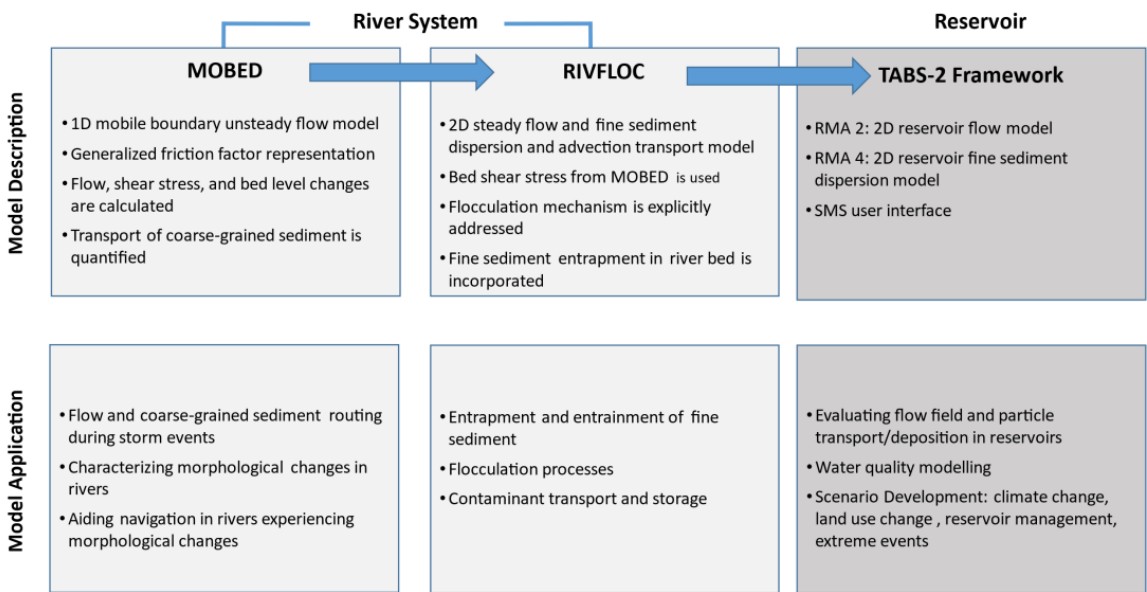

**Figure 2.** Schematic of the modelling framework.

### 2.2.2. Fine Sediment Transport Model—RIVFLOC

RIVFLOC is a cohesive sediment transport model which explicitly describes flocculation of fine sediment in the water column (Krishnappan [57]). It consists of two modules: a transport module and a flocculation module. The transport module is based on an advection-diffusion equation expressed in a curvilinear co-ordinate system to treat the transport and mixing of fine sediment entering a river as a steady source. The flocculation module incorporates a coagulation equation, which expresses the mass balance of the sediment during flocculation. Collision mechanisms such as Brownian motion, turbulent fluid shear, inertia of particles and differential settling are considered. The model uses an empirical relationship to express floc density as a function of floc size and a modified Stokes equation to calculate the settling velocity of solids in suspension. Specific details of the model and its application can be found in Krishnappan and Marsalek [60] and Droppo and Krishnappan [61].

### 2.2.3. TABS-MD with SMS User Interface

TABS-MD with SMS user interface consists of two models: RMA2 is a two-dimensional depth-averaged finite-element hydrodynamic model and RMA4 is a two-dimensional depth-averaged finite-element sediment transport and water quality model. Details of both models are reported in Donnell et al. [58] and Letter et al. [59], respectively. RMA2 and RMA4 are fully integrated by a user interface called SMS which is a graphical pre-and post-processor for numerical surface water models to allow interactive editing and display of finite element networks. Display controls allow the user to adjust color and line contouring to display either bed elevations or model results such as velocity fields and water surface elevations. SMS consists of a data module that includes tools for performing data analysis and interpretation. A mapping module allows the user to create a conceptual model and use background images to interface with the finite element mesh of the computational domain. A mesh module allows the creation of finite element meshes for different hydrodynamic modelling systems. SMS version 11 was used in this study.

### 2.2.4. Input Data Requirements for the Modelling Framework

The input data requirements for the component models of the modelling framework are shown in Table 1.

**Table 1.** Model input data requirements and corresponding sources.

| Component Models | Data Requirements | Data Sources |
|---|---|---|
| MOBED model | Hydraulic geometry and surface water elevation at 2 km intervals along the study reach for each river | 2011 cross sectional surveys described in Section 2.2.5 |
| | Bed material size data | |
| | Flow rate | Water survey of Canada Hydrometric stations for Crownsnest River @ Frank Stn 05AA008; for Castle River @ Ranger Station Stn 05AA028; and for Oldman River @ Range Road Stn 05AA035 |
| | Frictional parameters | Calibration of MOBED model described in Section 2.2.6. |
| RIVFLOC model | River geometry data: cross sectional shapes at a number of sections along the river | 2011 Cross sectional survey described in Section 2.2.5 |
| | Particle size distribution at the upstream boundary of the modelling domain | 2015 survey in the upper Crowsnest River (LISST measurements) described in Section 2.2.7. |
| | Suspended sediment concentration at the upstream boundary of the modelling domain | 2015 survey in the upper Crowsnest River described in Section 2.2.7 |
| | Relationship between the floc size and floc density | 2015 survey in the upper Crowsnest River described in Section 2.2.7. |
| | Bed shear stress distribution in the modelling domain | Provided by the MOBED model predictions |
| | Critical shear stress for deposition of fine sediment | Based on erosion and deposition experiments in annular flume Stone et al. [62] |
| | Cohesion parameter, β | Calibration parameter for RIVFLOC model-described in Section 2.2.8 |
| RMA2 model | Bathymetry data to formulate the finite element mesh | Provided by existing reservoir bathymetric data |
| | Flow rate at the upstream boundary of the reservoir | Provided by the output of the RIVFLOC model |
| RMA4 model | Two dimensional lateral velocity distribution in the reservoir | Provided by the output of the RMA2 model |
| | Suspended sediment concentration at the upstream boundary of the reservoir | Provided by the output of the RIVFLOC model |
| | Size distribution of the suspended sediment entering the reservoir at the upstream boundary of the reservoir | Provided by the output of the RIVFLOC model |

2.2.5. Cross-Section Survey of the Crowsnest, Castle and Oldman Rivers

In 2011, cross-section elevation and river-bed cobble size surveys were conducted at ~2 km intervals along 54, 46 and 46 km reaches of the Crowsnest, Castle and Oldman Rivers, respectively (Figure 1). Hydrometric data collected at Water Survey of Canada gauging stations (Crowsnest River, 05AA008; Castle River, 05AA028; Oldman River, 05AA035) were used to set the boundary conditions for the models. The physical characteristics (slope, width and bed materials) of the Castle, Crowsnest and Oldman study reaches are variable, and the river-bed elevation decreased by ~250 m over a distance of ~50 km. The average slope of the Crowsnest, Castle, and Oldman Rivers is 0.44%, 0.43% and 0.46%, respectively. The Crowsnest River is narrower (10 to 20 m wide) than the Castle and Oldman Rivers which can vary in width from 70 to 100 m. Bed materials in these rivers consist primarily of coarse sand, pebbles and cobbles. River-beds are armoured and stable during low flows but mobilisation and transport of bed material can occur during flood stages. Photographs of the bed materials of the three river sections are shown in Figure 3.

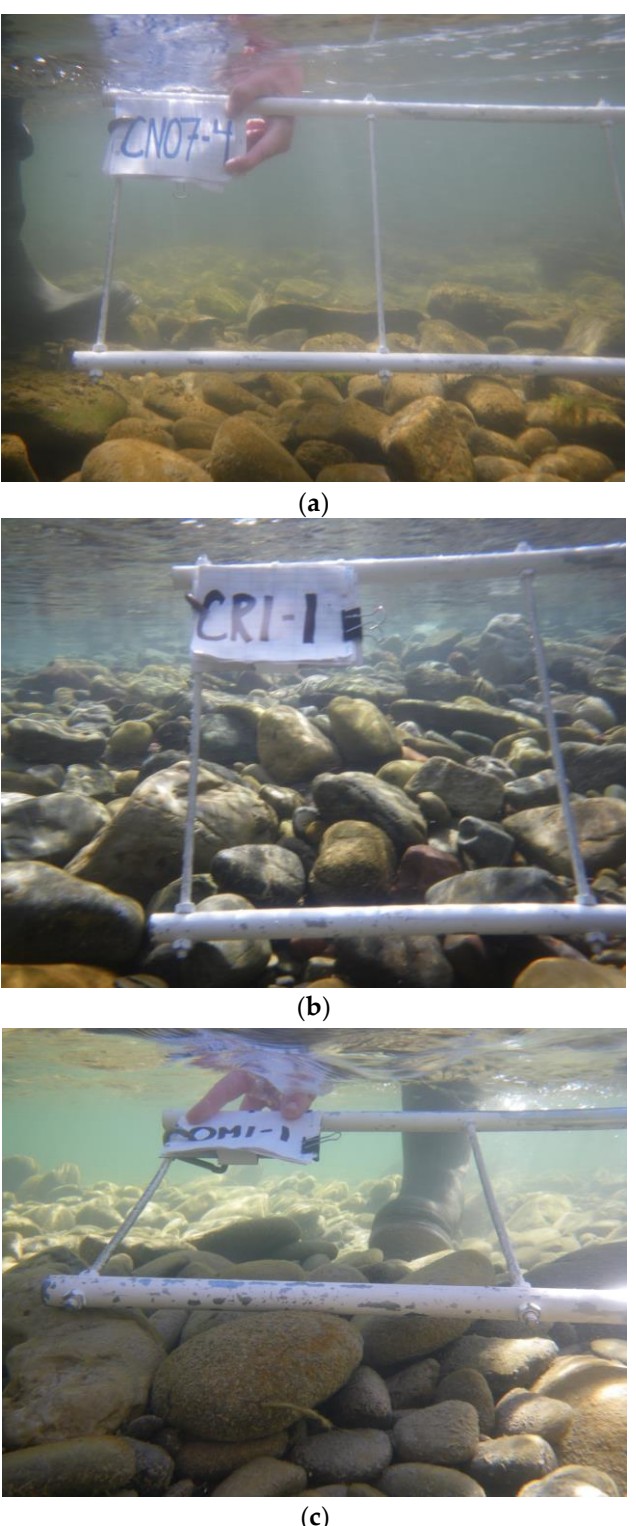

**Figure 3.** Sample photographs of river-beds: (**a**) Crowsnest River, (**b**) Castle River and (**c**) Oldman River.

2.2.6. Calibration of MOBED

Using the cross-sectional shape and the river profile data, the MOBED model was calibrated for the flow conditions that existed during the cross-section shape surveys. At the upstream boundary, a flow rate boundary condition was used and, at the downstream boundary, a stage–discharge relationship derived from the measured data was used. Bed material size at each cross-section was estimated from underwater photographs (using

a metal frame grid which measures 25 cm × 25 cm) of the bed material taken at five equidistant locations across each section. An image analysis system was used to calculate the bed material size distribution characteristics such as $D_{35}$, $D_{50}$, and $D_{65}$ needed for the MOBED model. Calibration of the MOBED model was completed by running the model for flow rates that existed when the cross-section surveys were carried out and the calculated water surface elevations were compared with the measured data. A Manning roughness coefficient that produced the best match between the predicted and measured data was determined. Comparisons of the measured and predicted water surface elevations for all three river reaches are shown in Figure 4. A Manning's roughness parameter of 0.070 was estimated for all of the three study reaches and this value is comparable to previously reported values for similar sized bed materials (Hey, et al. [63]). The accuracy of calibration was estimated by comparing the predicted water surface elevation and the measured water surface elevation and the percent deviation varied in the range of ±0.1 to ±1.0 percent of the measured water surface elevation. The MOBED model had been field verified by a number of earlier studies (Krishnappan [54–56]) and hence no attempt was made here to validate the model.

The bed shear stress predicted by the MOBED model for the three river reaches is shown in Figure 5. The magnitude of the bed shear stresses ranged from 10 to 50 Pa. The variability of the predicted bed shear stress is due to variation in river geometry (depth and width) governing mean flow characteristics.

2.2.7. Fine Sediment Transport Survey in 2015 in the Upper Crowsnest River

In 2015, a detailed survey of suspended sediment concentrations and effective particle size distributions was conducted in situ for two different flow conditions in the upper Crowsnest River over a reach of about 20 km. Effective particle size distributions were measured using a laser particle size analyzer (LISST 100X; Sequoia Scientific, Bellevue, WA, USA) at several locations near the confluences of three tributary inflows to the Crowsnest River including Star Creek (1035 ha, mean slope = 27%, undisturbed), South York Creek (365 ha, mean slope = 56%, 54% burned) and Lyons Creek (1309 ha, mean slope = 46%, 100% burned and 20% salvage logged). At each confluence, discharge, effective particle size distributions and suspended sediment concentrations were measured in, above and below the confluence of each tributary with the Crowsnest River (see Figure 6).

This measurement campaign provided the input data necessary for setting up the RIVFLOC model (see Table 1). Using these data (flow rate, suspended sediment concentration and the effective particle size distribution of fine sediment at the upstream boundary of the modelling domain), the values of bed shear stresses predicted by MOBED and the critical shear stress for the deposition of the sediment given by the laboratory investigation by Stone et al. [62], the RIVFLOC model was applied to the upper 20 km stretch of the Crowsnest River and the cohesion parameter, β was established as part of the calibration procedure. Details of this calibration procedure are described in Section 2.2.8.

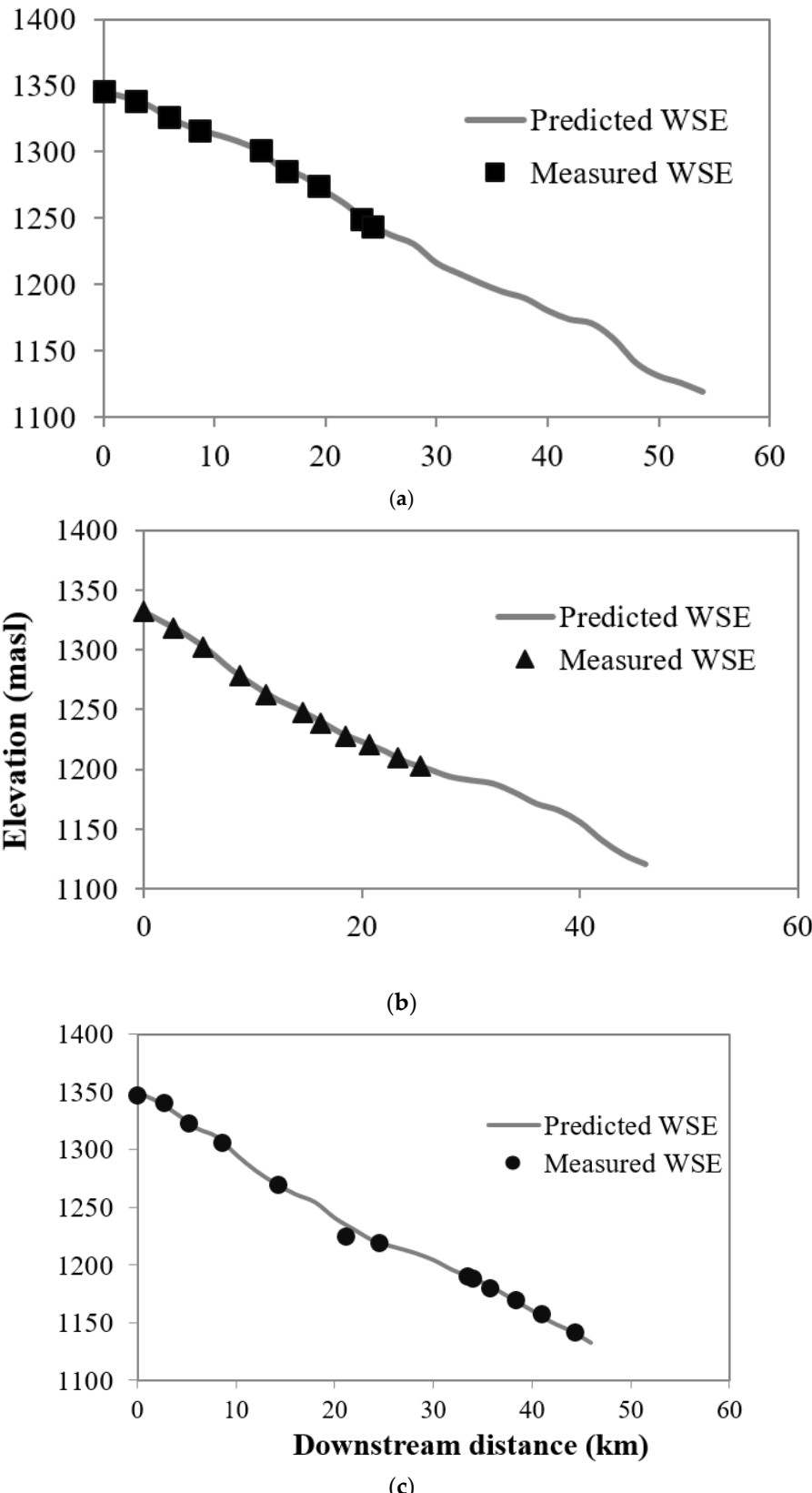

**Figure 4.** Comparisons between measured and predicted water surface elevations for the Crowsnest River reach ((**a**) Flow rate = 2.65 m$^3$/s; Manning's n = 0.070), Castle River reach ((**b**) Flow rate = 4.0 m$^3$/s; Manning's n = 0.070) and Oldman River reach ((**c**) Flow rate = 5.0 m$^3$/s; Manning's n = 0.070).

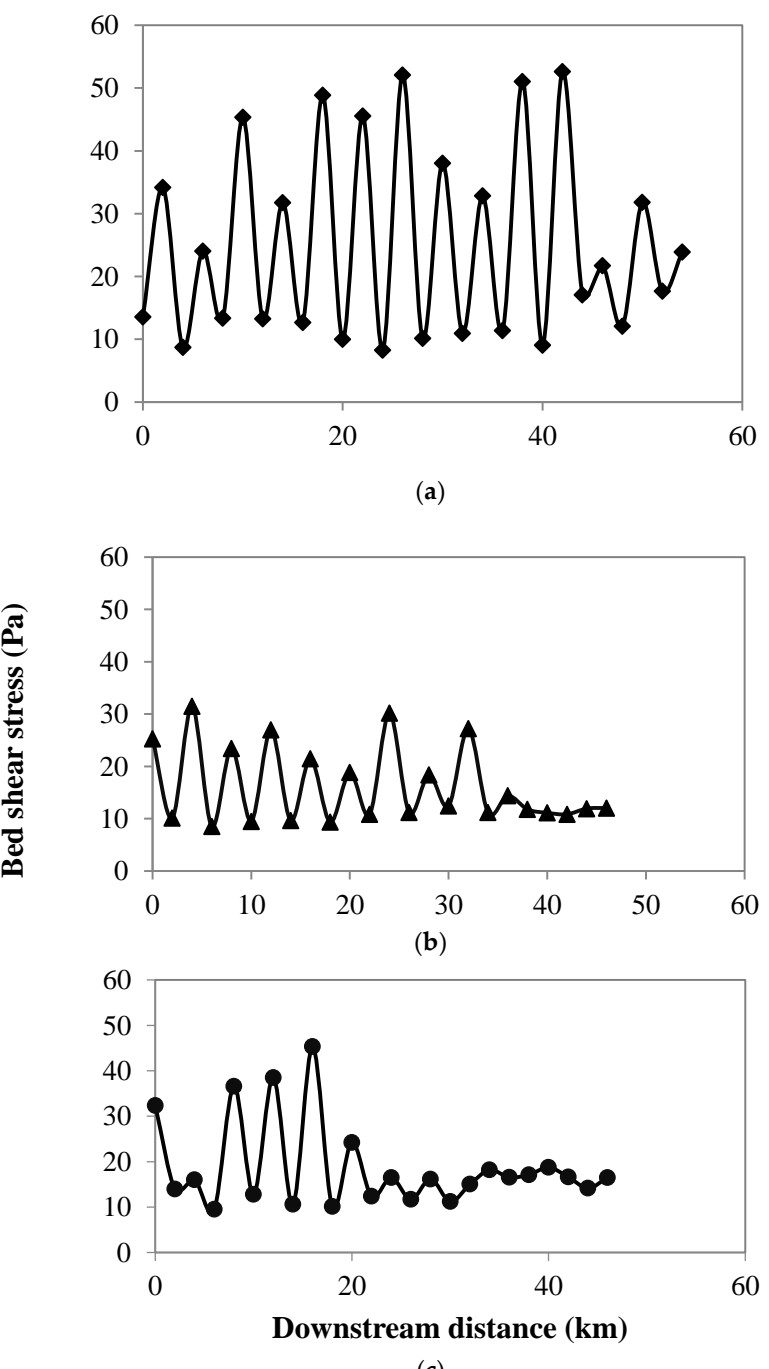

**Figure 5.** Variation of predicted bed shear stress with distance along the river reaches: Crowsnest River ((**a**) Flow rate = 2.65 m³/s; Manning's n = 0.070), Castle River ((**b**) Flow rate = 4 m³/s, Manning's n = 0.070), Oldman River ((**c**) Flow rate = 5 m³/s, Manning's n = 0.070).

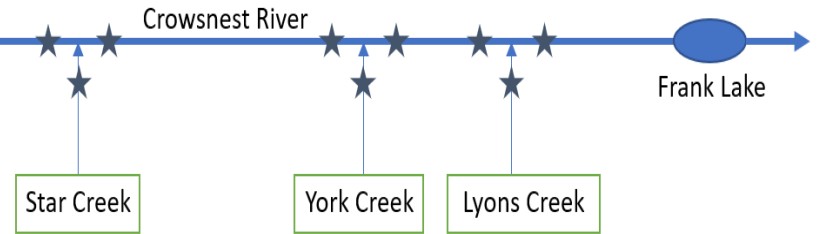

**Figure 6.** A schematic diagram of the sampling sites in the upper Crowsnest River.

### 2.2.8. Calibration of RIVFLOC

Hydrometric and sediment survey campaigns were carried out, in May and June 2015. The flow rate and suspended sediment concentration data are summarized in Tables 2 and 3. In May 2015, sediment concentrations were lower in the tributaries than the main stem of the Crowsnest River. Mixing occurred at the confluence of tributary inflows with the main channel and concentrations of suspended sediment decreased due to a dilution effect (Table 2). Calculation of sediment concentrations downstream of the confluence with the modelling framework, assuming complete mixing and no interaction of the sediment with the bed, yielded values that agree favourably with the measured concentrations (see measured and predicted values in Table 2). The contribution of sediment to the main channel from tributary inflow accounted for 1.0 to 3.3% of the sediment mass in the Crowsnest River. In June 2015, downstream trends in suspended sediment were similar to those observed in May, except that flow rates and sediment concentrations were much higher (Table 3). At this time, the contribution of sediment loading from the tributaries ranged from 3 to 8%.

Representative effective particle size distributions of suspended sediment measured directly in the water column for two stations (upstream of Star Creek confluence and downstream of Lyons Creek confluence) in the Crowsnest River are shown in Figure 7. Particle size distributions at these two stations were similar for smaller particle size ranges, but the distributions deviate from each other as particle size increases. For particles > 200 μm, the distribution corresponding to the station above Star Creek was slightly smaller in comparison to the distribution corresponding to the station below Lyons Creek. This deviation is due to flocculation of suspended solids in the water column. Evidence of flocculation is provided by direct observation using sediment collected on a filter paper and observed using an inverted microscope (Figure 8). A representative photomicrograph confirms the presence of flocs in the sediment population.

**Table 2.** Summary of suspended sediment measurements in the upper Crowsnest River (CNR) and tributaries (May 2015). Study sites relate to those shown in the schematic in Figure 6.

| Study Site | Distance (m) | Measured Discharge (m$^3$/s) | Estimated Discharge (m$^3$/s) | Measured Concentration (cc/m$^3$) | Predicted Concentration (cc/m$^3$) | % Difference | Sed Load (cc/s) | % Contribution |
|---|---|---|---|---|---|---|---|---|
| CNR-u/s Star | 5000 | 4.97 | | 23.3 | | | 116.4 | |
| Star | 5500 | 0.31 | | 6.0 | | | 1.8 | 1 |
| CNR-d/s Star | 6000 | | 5.28 | 23.0 | 22.4 | 2.4 | 118.2 | |
| CNR-u/s York | 14,000 | 6.9 | | 25.6 | | | 176.7 | |
| York | 14,500 | 0.82 | | 7.7 | | | 6.3 | 3.3 |
| CNR-d/s York | 15,000 | | 7.72 | | 23.7 | | 183.0 | |
| CNR-u/s Lyons | 16,000 | 7.45 | | 25.2 | | | 188.0 | |
| Lyons | 16,500 | 0.51 | | 6.2 | | | 3.2 | 1.7 |
| CNR-d/s Lyons | 17,000 | | 7.96 | 22.8 | 22.6 | 0.6 | 191.2 | |
| Frank Lake | 20,000 | 8.45 | | | | | | |

**Table 3.** Summary of suspended sediment measurements in the upper Crowsnest River (CNR) and tributaries (June 2015). Study sites relate to those shown in the schematic in Figure 6.

| Study Site | Downstream Distance in m | Measured Discharge (m³/s) | Estimated Discharge (m³/s) | Measured Concentration (cc/m³) | Predicted Concentration (cc/m³) | % Difference | Sed Load (cc/s) | % Contribution |
|---|---|---|---|---|---|---|---|---|
| CNR-u/s Star | 5000 | 13.3 | | 33.1 | | | 444 | |
| Star | 5500 | 1.1 | | 32.5 | | | 34.9 | 3.1 |
| CNR-d/s Star | 6000 | | 14.4 | 32.5 | 33.1 | 1.8 | 475 | |
| CNR-u/s York | 14,000 | | 20.6 | 51.7 | | | 1066 | |
| York Creek | 14,500 | 2.7 | | 22.8 | | | 62.0 | 5.5 |
| CNR-d/s York | 15,000 | | 23.4 | 46.7 | 48.3 | 3.4 | 1129 | |
| CNR-u/s Lyons | 16,000 | | 23.4 | 44.9 | | | 1094 | |
| Lyons | 16,500 | 2.4 | | 35.8 | | | 85.9 | 7.6 |
| CNR-d/s Lyons | 17,000 | | 25.8 | 46.3 | 44.1 | 4.8 | 1135 | |
| Frank Lake | 20,000 | | | | | | | |

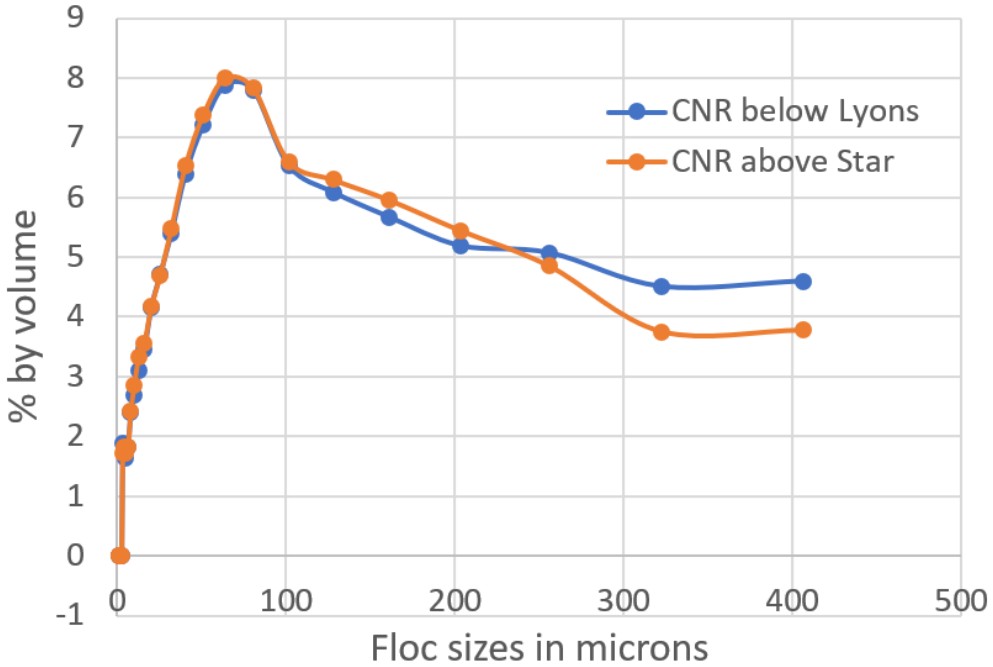

**Figure 7.** Effective particle size distributions of suspended sediment at two stations in the upper Crowsnest River.

The flocculation module of RIVFLOC requires data on the density and settling velocity of flocs which are dependent on floc size (Lau and Krishnappan [64]). The relationship between floc density and floc size was determined by utilizing simultaneous measurements of volumetric concentration of suspended sediment as measured by the LISST 100X during the 2015 sediment survey and the mass concentration of suspended sediment samples collected during the same survey. The relationship between size, density and settling velocity of flocs is presented in Figure 9.

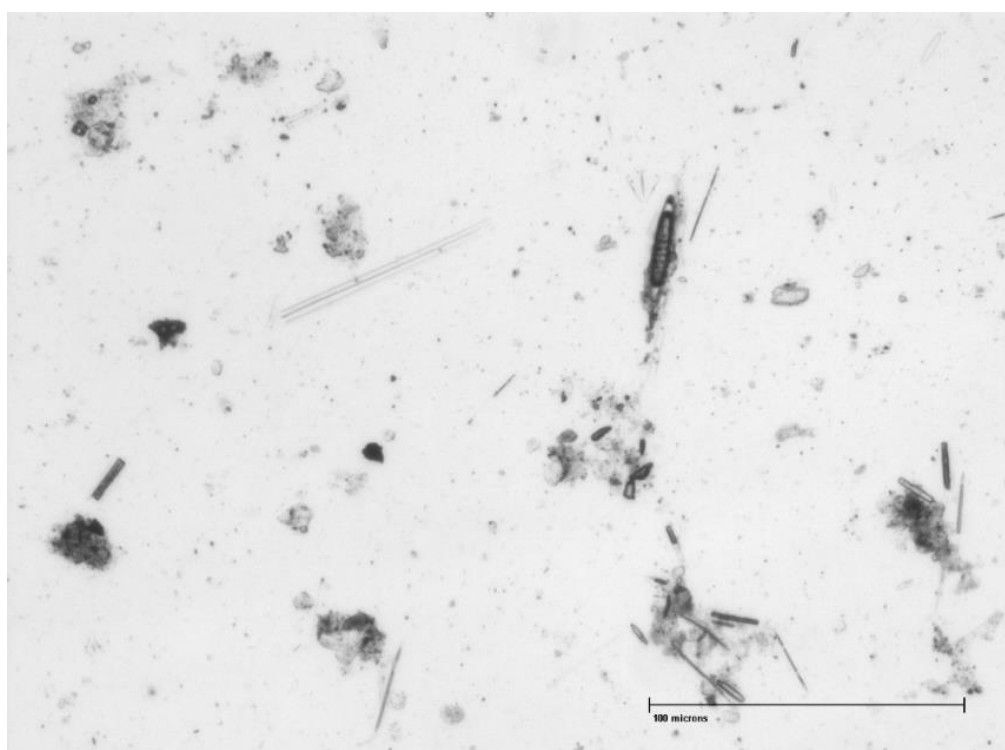

**Figure 8.** Photomicrograph of the suspended sediment in the Crowsnest River.

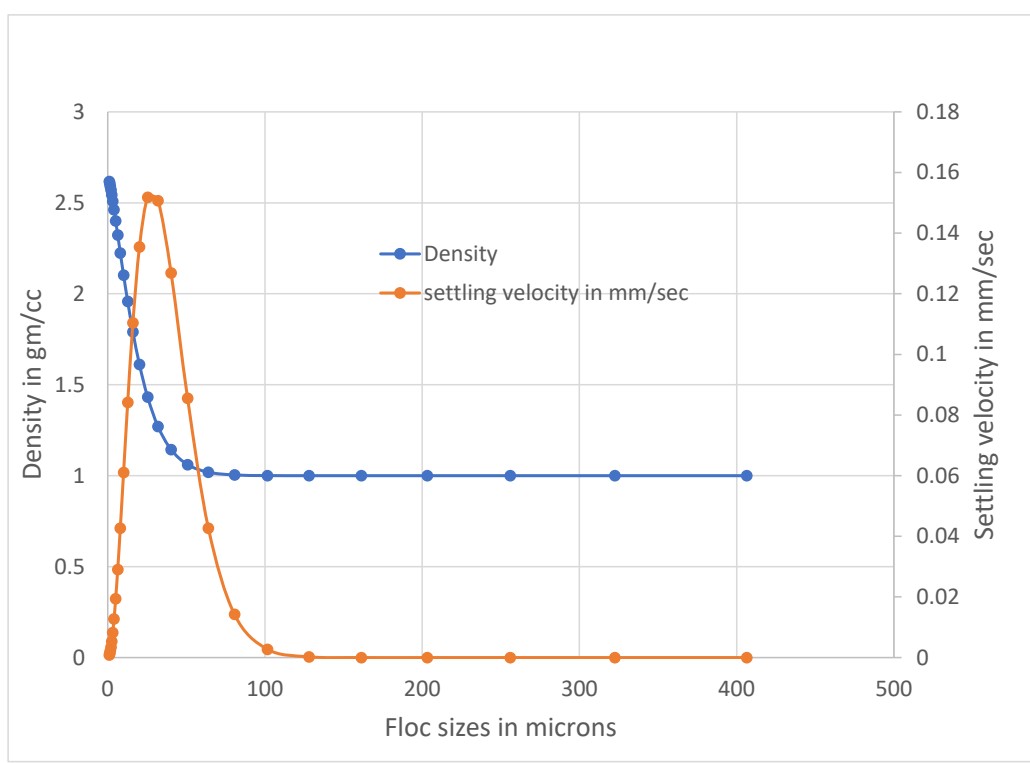

**Figure 9.** Density and settling velocity of sediment flocs as a function of floc sizes for suspended sediment in the Crowsnest River.

Because water is incorporated into the assemblage of particles in larger flocs, these data show that floc density decreases with increasing floc size (Droppo [65]; Krishnappan and Stephens [66]). The relationship between decreasing density as a function of floc size

results in a complex floc settling behaviour. For smaller floc sizes, the settling velocity increases as a function of floc size and reaches a maximum value but with further increases in floc size, the settling velocity decreases and approaches a value close to zero for larger flocs. This settling behaviour of velocity vs floc size has also been observed in previous studies (Droppo [65]; Krishnappan [39]).

In RIVFLOC, the sediment flux ($q_{sd}$) at the sediment-water interface is specified using the Krone's [34] formulation. According to this formulation, $q_{sd}$ is given by

$$q_{sd} = p w_s C_b \tag{1}$$

where $p$ is a measure of the probability that a floc, settling to the bed, stays in the bed. Krone proposed a relationship for $p$ as:

$$p = \left(1 - \frac{\tau}{\tau_{crd}}\right) \text{ for } \tau < \tau_{crd} \tag{2}$$

$$p = 0 \text{ for } \tau > \tau_{crd} \tag{3}$$

where $\tau$ is the bed shear stress and $\tau_{crd}$ is the critical shear stress for deposition, which is defined as the bed shear stress above which none of the initially suspended sediment would deposit.

For applying the RIVFLOC model to the upper Crowsnest River reach, the upstream boundary conditions were specified using the distribution of volumetric sediment concentration and size distribution measured at the Crowsnest River station located upstream of the confluence with Star Creek. To specify the boundary condition at the sediment-water interface, the bed shear stress and the critical shear stress for deposition of the sediment are required, as previously described. Bed shear stresses for the flow field were computed using MOBED. The critical shear stresses for the deposition of sediment were obtained from a laboratory flume study carried out by Stone et al. [62]). The measured critical shear stress for erosion ($\tau_{crit}$) of burned and un-burned cohesive sediments from streams in the Castle River watershed (Figure 1) for different consolidation and bio-stabilization conditions (2, 7, 14 days) are presented in Table 4. The critical shear stresses thresholds for deposition of suspended solids in the Crowsnest River were deduced from the values shown in Table 4 using the results from earlier laboratory studies on cohesive sediments from different sources (Krishnappan and Stephens [66]; Krishnappan [38]; Krishnappan et al. [67]). Previous laboratory studies on cohesive sediments demonstrate that the critical shear stress for deposition (i.e., the shear stress below which all of the initially suspended sediment would deposit) is about one half of the value for the critical shear stress for erosion (Partheniades [33]). These studies show that the critical shear stress for deposition, as defined by Partheniades [33]), which is the shear stress below which all of the initially suspended sediment would deposit, is about one tenth of the critical shear stress for deposition as defined by Krone [34], i.e., the shear stress above which none of the sediment in suspension would deposit. Using these two results, the critical shear stresses for deposition as defined by Krone [34]), which are needed for the RIVFLOC model were calculated by multiplying the values in Table 4 by five and the resulting values are listed in Table 5. Since the bed shear stress ($\tau$) predicted by MOBED (10 to 50 Pa) is considerably larger than the critical shear stress for deposition (Table 5), the depositional flux becomes zero and under this boundary condition, RIVFLOC predicts a constant fine sediment concentration as a function of distance along the study reach, suggesting that the fine sediment is simply propagated through the river channel system.

The cohesion parameter, β required for RIVFLOC was determined by applying the model to a sub-reach of the Crowsnest River between the confluences of Star Creek and Lyons Creek. An effective particle size distribution measured at a cross-section above the confluence of Star Creek using a LISST 100X during the 2015 sediment survey was used as the upstream boundary condition. The size distribution of sediment flocs for the station downstream of Lyons Creek was predicted using RIVFLOC for various values of the

cohesion parameter, β. The predictions were compared with measured size distributions for various values of β, and a value of 0.075 was chosen as an acceptable value. As can be seen from Figure 10, there is a reasonable agreement between the measured and predicted distributions for both smaller and larger flocs. However, for flocs in the intermediate size range (100 to 200 μm), the model over predicts the measured values somewhat. The percent deviation of the predicted floc size and the measured value ranged from ±1.0% to ±10%. The value of 0.075 for β is in the range of values that were measured in other similar studies that were carried out for sediments in Alberta (Droppo and Krishnappan [61]).

**Table 4.** Critical shear stresses for erosion and settling velocity of fine sediment from the Castle River watershed (Source: Stone et al. [62]).

| Sediment Type | Consolidation Period (Days) | Settling Velocity mm/s | Critical Shear Stress for Erosion in Pa |
|---|---|---|---|
| Burned | 2 | 2.2 | 0.08 |
| | 7 | 2.8 | 0.16 |
| | 14 | 3.0 | 0.18 |
| Unburned | 2 | 3.2 | 0.04 |
| | 7 | 3.3 | 0.10 |
| | 14 | 3.8 | 0.09 |

**Table 5.** Critical shear stresses for deposition of sediment (Krone's definition).

| Sediment Type | Consolidation Period (Days) | Settling Velocity mm/s | Critical Shear Stress for Deposition in Pa |
|---|---|---|---|
| Burned | 2 | 2.2 | 0.40 |
| | 7 | 2.8 | 0.64 |
| | 14 | 3.0 | 0.72 |
| Unburned | 2 | 3.2 | 0.20 |
| | 7 | 3.3 | 0.50 |
| | 14 | 3.8 | 0.45 |

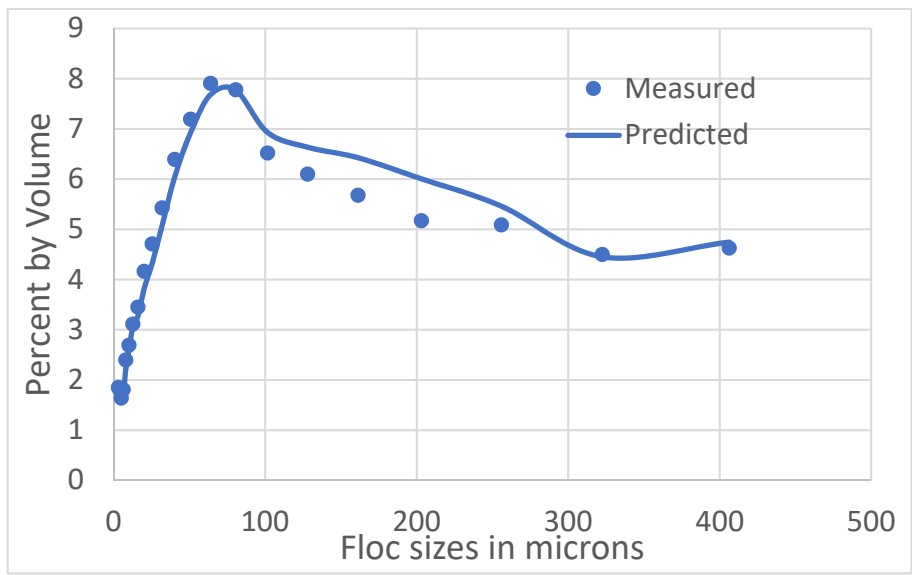

**Figure 10.** Comparison between measured size distribution and RIVFLOC prediction for the station downstream of the Lyons Creek confluence on the Crowsnest River.

### 2.2.9. Setting Up of TABS-MD for the Oldman Reservoir

TABS-MD (RMA2 and RMA4 with SMS user interface) was used to create a numerical grid based on reservoir bathymetry and to simulate both the flow field as well as fine sediment transport and dispersion within the Oldman Reservoir. To predict the flow pattern in the reservoir, an example stream flow condition of 25 m$^3$/s in the Crowsnest River, 75 m$^3$/s from the Oldman River and 100 m$^3$/s from the Castle River was used. These higher-than-average inflow rates were used to produce a sizable flow field that was expected to carry a significant sediment flux into the reservoir. The Manning's roughness coefficient for the reservoir was set as 0.025 (Hey [63]) and the turbulent eddy viscosity coefficient for the model was specified in terms of a Peclet Number with a value equal to 20 (Donnell et al. [58]). The flow patterns and velocity magnitude contours predicted by the model illustrate the complex flow patterns in the reservoir which are due to the irregular morphology and bathymetry. The prediction of sediment propagation over a 16-day period was evaluated using the model and is shown in Figure 11 to illustrate its capability. In this example, a sediment concentration of 1000 mg/L was used at the outlet of all three rivers. The model correctly predicts the flow rate through the dam thereby preserving flow continuity. Velocity in the reservoir ranged from 1 to 3 cm/s. Bed shear stresses predicted by the model range from 0.001 to 0.003 Pa and are two orders of magnitude lower than the critical shear stress for deposition of sediment shown in Table 5. Accordingly, the model predicts that most of the fine sediment entering the reservoir from all three rivers will be deposited within the reservoir and that resuspension of bottom sediment at normal operating water levels is unlikely.

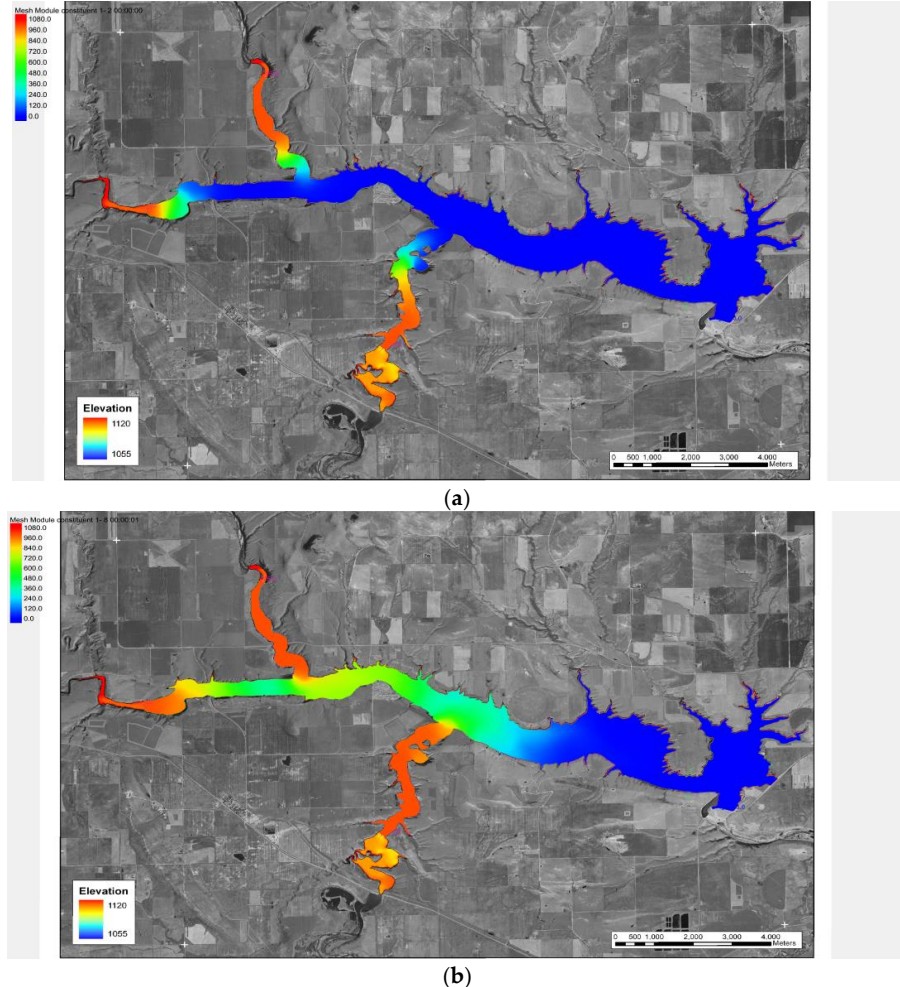

(a)

(b)

**Figure 11.** *Cont.*

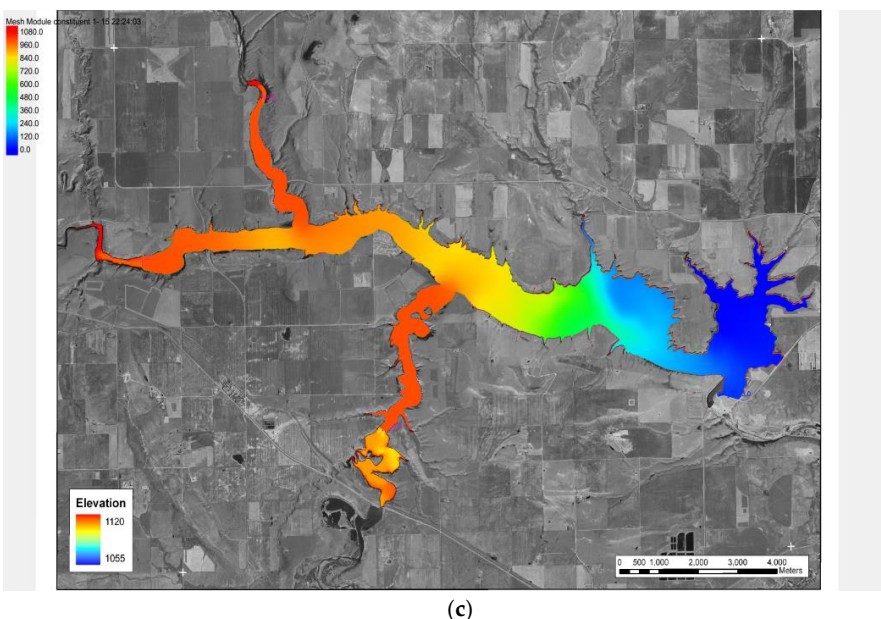

(**c**)

**Figure 11.** Simulation of sediment propagation through the Oldman Reservoir as predicted by TABS-MD ((**a**) Elapsed time = 2 days; (**b**) Elapsed time = 8 days; (**c**) Elapsed time = 16 days). The elevation legends shown as insets in these figures need to be ignored.

## 3. Results

The calibrated RIVFLOC model was run to estimate sediment flux for the Crowsnest River tributary to the Oldman reservoir (Figure 11). Predictions from RIVFLOC show that fine sediment in the Crowsnest River is transported directly to the Oldman reservoir if sediment entrapment in gravel beds is assumed to be zero (top horizontal line in Figure 12). However, several previous laboratory studies demonstrate that fine sediment deposition can occur in gravel beds because of the entrapment process (Einstein [68]; Packman et al. [69]; Rehg et al. [70]; Krishnappan and Engel [71]). The effect of coarse gravel on cohesive sediment entrapment was evaluated in an annular flume using fine sediment and gravel from the Elbow River in Alberta, which has similar land use characteristics (geology, vegetation, land use) to the Crowsnest River (Glasbergen et al. [72]). To evaluate the effect of entrapment on fine sediment transport dynamics in the Crowsnest River, two entrapment coefficients chosen from the literature (Krishnappan and Engel [71]; Glasbergen et al. [72]) were applied to RIVFLOC. The results show that the amount of sediment transported in the Crowsnest River decreases with distance downstream as the entrapment coefficient increases (Figure 12). Detailed field studies are required to quantify sediment entrapment dynamics in the Crowsnest River to refine model predictions. However, an entrapment coefficient (ratio between the entrapment flux and the settling flux) of 0.20 determined by Glasbergen et al. [72] provides a reasonable value and is used in this study to model fine sediment transport to the Oldman Reservoir. With this value of the entrapment coefficient, RIVFLOC predicts that 22% of the sediment entering the Crowsnest River will be entrapped within the river reach whereas the remaining 78% will be delivered downstream to the Oldman Reservoir.

RIVFLOC was also used to calculate the effective size distribution of solids at the downstream boundary of the Crowsnest River. Figure 13 compares this prediction with the size distribution of the sediment entering the river at the upstream boundary. From Figure 13, we can see that the effective size distribution at the downstream boundary increased with distance downstream due to flocculation. The predominant floc size at the upstream boundary was 70 μm but increased by a factor of 1.7 (~120 μm) at the downstream boundary. The effective sediment size distribution calculated from RIVFLOC

for the downstream boundary, was used as the input for sediment size distribution for the reservoir model (RMA4).

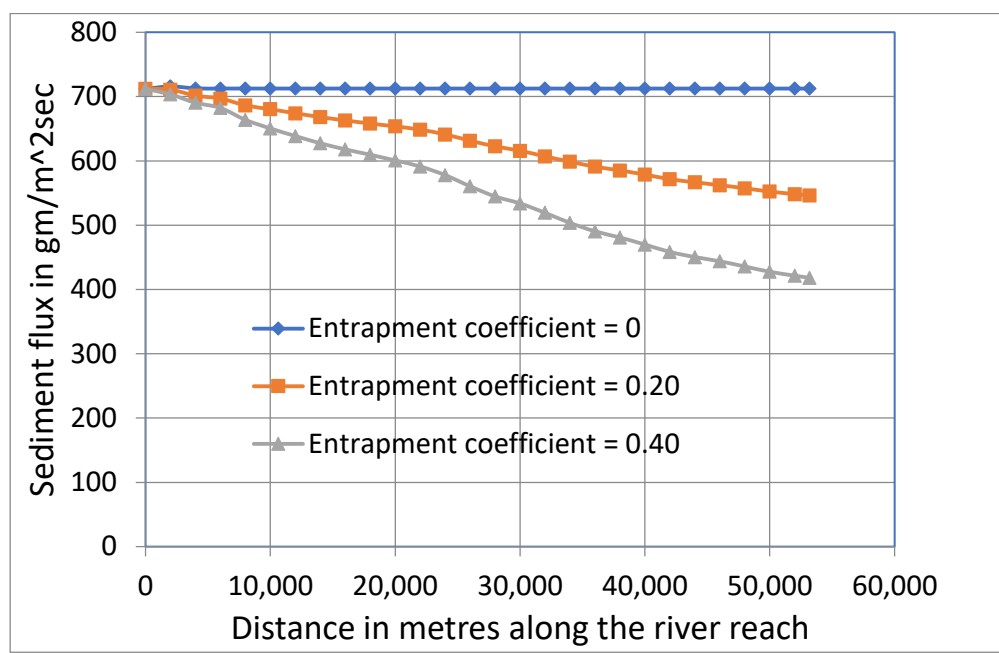

**Figure 12.** Fine sediment transport rate computed by RIVFLOC for the full Crowsnest River reach.

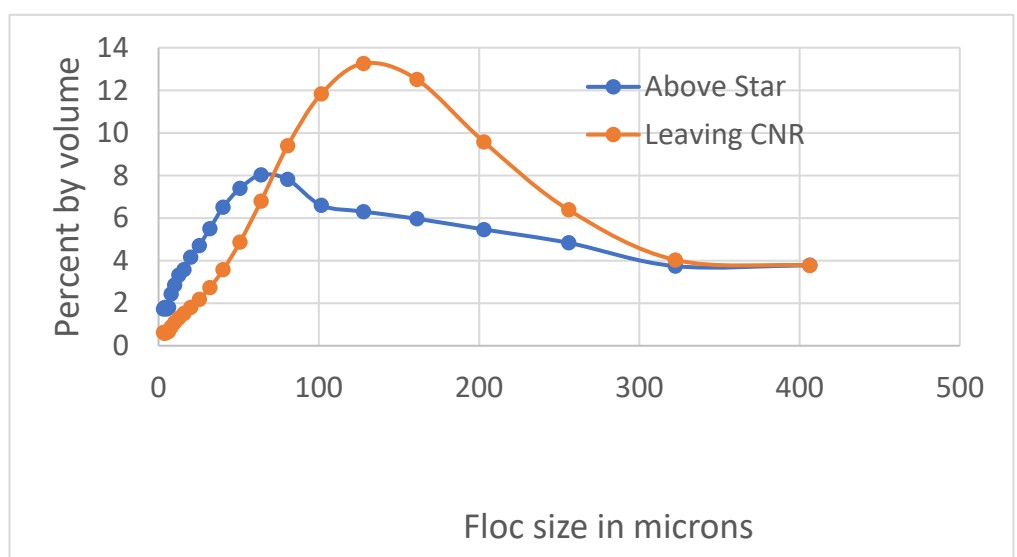

**Figure 13.** Comparison of effective particle size distributions of sediment entering and leaving the Crowsnest River reach.

RMA4 was used to determine the concentration of suspended sediment for individual size classes in the river inflow and at three transects (A, B, C) across the Oldman reservoir (Figure 1). Sediment deposition is simulated in RMA4 using the decay rate coefficient (k) which is equal to settling velocity divided by an average depth for any sediment size class. Simulated changes in sediment concentrations for various size fractions in different parts of the reservoir are presented in Figure 14. For this simulation, sediment inputs at the upstream boundary were held constant but different upstream boundary conditions in real time can also be used for scenario development or real time simulations. Previous research has demonstrated that the settling of flocs occurs at different rates for different

size classes (Krishnappan [39]). RMA4 was used to simulate spatial variation in effective sediment size and concentration at the upstream, middle, and downstream sections of the reservoir (Figure 14). The data show that sediment in the size classes around 30 μm settles relatively quickly while both the finer (<30 μm) and larger sediment flocs (>100 μm) remain in suspension because of lower settling velocities. Given that bed shear stresses in the reservoir are two orders of magnitude lower than the critical shear stress for sediment deposition, the majority of suspended sediment entering the reservoir will deposit. The only size classes that are likely to be carried with the flow downstream of the Oldman dam are those that stay in the water column due to Brownian motion and the larger flocs that have very low settling velocities.

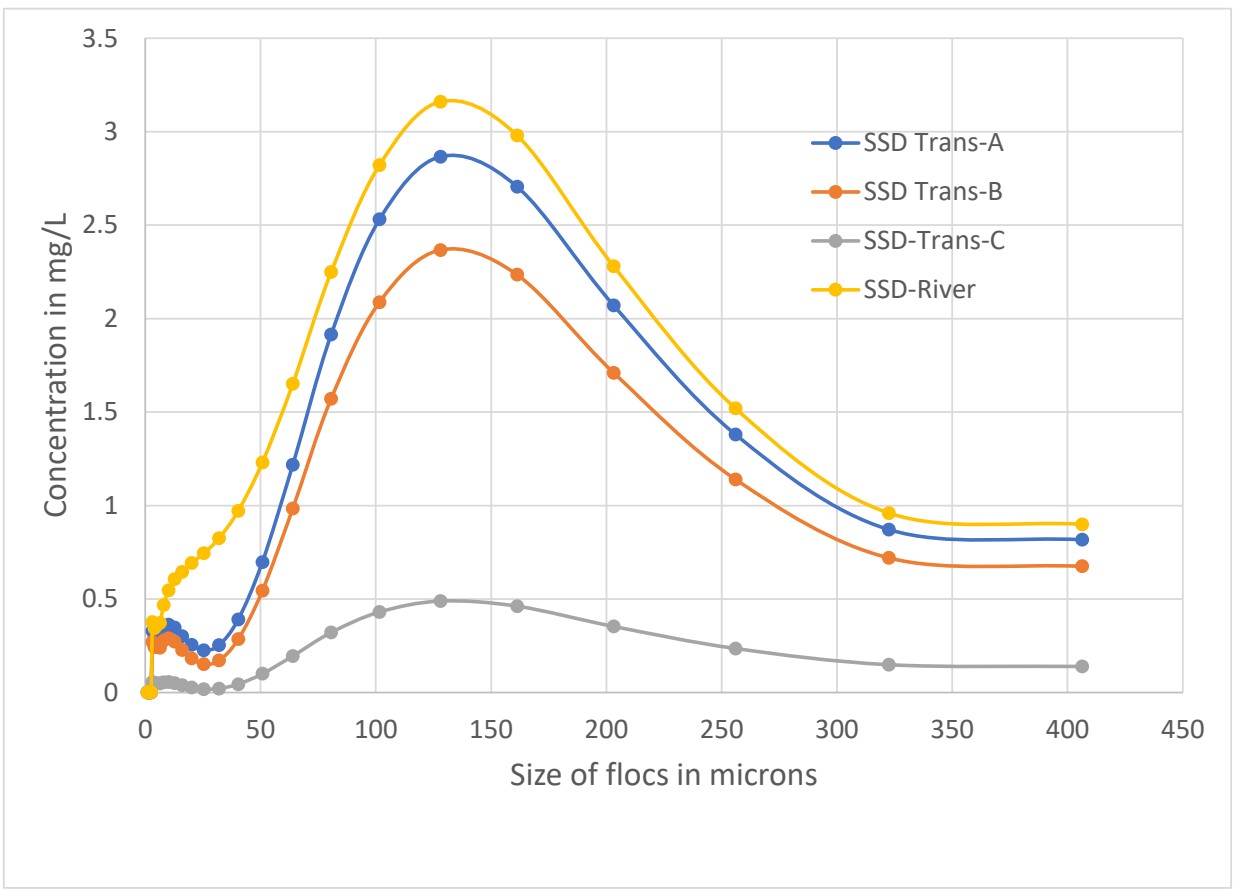

**Figure 14.** Effective particle size distributions of suspended sediment in river inflow and in the three transects across the Oldman Reservoir.

*Simulating Sediment Flux to the Oldman Reservoir from Upstream Tributary Inflows to the Crowsnest River*

While significant advancements in cohesive sediment transport models have been made, they are based primarily on the results of laboratory studies and are very seldom verified under field conditions particularly at large basin scales (Willis and Krishnappan [73]). One of the primary applications for the generic modelling framework described herein, is its utility for scenario development to explore specific risks of disturbance impacts (i.e., wildfire, extreme events, harvesting) on the propagation and fate of fine sediment in a reservoir from upstream sources. Here, we incorporate a five-year data set consisting of mean daily streamflow and suspended sediment concentration data as input in the modelling framework to simulate sediment transport dynamics in source water forested regions impacted by wildfire, simulating both propagation to, and through and, deposition within, a reservoir (Table 6). Simulations were made using an entrapment coefficient

of 0.2 (Glasbergen et al. [72]) and results from the model are used to compare the amount and fate of suspended sediment from burned and post fire salvage logged watersheds (Lyons Creek) with that generated from predominantly unburned forested watersheds (Star Creek) and partially burned (South York Creek).

**Table 6.** Suspended sediment export (metric tonnes per year) from three Crowsnest River tributaries.

| Tributaries | Sed. Type | 2005 | 2006 | 2007 | 2008 | 2009 | Total |
|---|---|---|---|---|---|---|---|
| Star Creek | unburned | 3.26 | 26.66 | 12.98 | 14.38 | 5.91 | 63 |
| South York | burned (56%) | 2.57 | 21.83 | 80.73 | 71.59 | 27.33 | 204 |
| Lyons Creek | burned | 298.7 | 16.71 | 19.38 | 1406.87 | 46.05 | 1788 |
| Total | | 304.7 | 65.2 | 113.09 | 1492.84 | 79.29 | 2055.11 |

The results of model simulations show differences in the depositional patterns for burned and unburned sediment deposited in four zones (Zone 1 is the region between the Crowsnest River entrance and the Transect A, Zone 2 is the region between Transect A and Transect B, Zone 3 is the region between the Transect B and Transect C and Zone 4 is the region between Transect C and the reservoir outlet) of the Oldman reservoir over the 5-year simulation period (Table 7).

**Table 7.** Sediment mass deposited in the Oldman Reservoir from burned and unburned tributary inflows.

| Reservoir Zone | Unburned Sediment | | Burned Sediment | |
|---|---|---|---|---|
| | Mass (t) | % | Mass (t) | % |
| 1 | 39.3 | 33 | 385.8 | 26.0 |
| 2 | 16.1 | 13.5 | 210.7 | 14.2 |
| 3 | 51.4 | 43.1 | 712.2 | 48.0 |
| 4 | 12.4 | 10.4 | 175.1 | 11.8 |
| Total | 119.2 | 100 | 1483.8 | 100 |

The modelled sediment deposition patterns predicted by RMA4 demonstrate how differences in settling characteristics of the two sediment types influence dispersion and fate in the reservoir. The distinction between the behaviour of burned and unburned sediment is primarily due to the differences in corresponding settling velocities and this distinction is manifested in the reservoir because of the lower shear stresses that exist in this receptor. In the river channels, however, both burned and unburned sediment behave similarly because of the dominance of the bed shear stress over the small differences in the critical conditions between the two sediment types. Sediment generated from Lyons Creek (1788 tonnes) and partially burned South York Creek (204 × 0.56 = 114.2 tonnes) accounted for 93% of total sediment production (2055 tonnes) from the three monitored tributary inflows (Table 6). Out of the total amount of sediment produced, about 80% of the sediment is deposited in the reservoir (Table 7). To examine the impact of the 2003 Lost Creek wildfire on regional scale sediment production, Stone et al. [26] used a composite geochemical fingerprinting procedure to apportion the sediment efflux from three key spatial sediment sources: (1) unburned (reference), (2) burned and (3) burned sub-basins that were subsequently salvage logged. They reported that >80% of the downstream sediment contribution to the Oldman reservoir was produced from ~14% of the upstream landscape that was affected by the wildfire. The results of this source apportionment study agree favourably with modelled estimates of spatial sediment loading generated in the present study. Accordingly, the modelling framework can be accepted as providing planning and management level estimates of sediment delivery to the Oldman reservoir.

## 4. Discussion

The transport of cohesive sediment in aquatic systems is characterized by interactions among fine-grained primary sediment particles that cause flocs to form (Droppo [65]). Flocs have relatively low densities, large pore spaces and reactive surfaces that remove contaminants from the water column (Krishnappan [74]). The flocculation mechanism is dependent upon several factors including particle mineralogy, electrochemical nature of the flowing medium, biological factors such as bacteria and presence of other organic material and hydrodynamic properties of the flow field (Droppo et al. [75]). In a study of river and lake sediment, Droppo et al. [76] reported that only flocs < 100 μm (equivalent spherical diameter) settled within the Stokes' region (Re < 0.2). The densities of these flocculated materials ranged from 1 to 1.4 g cm$^{-3}$ but the majority of flocs had densities of less than 1.1 g cm$^{-3}$. Floc porosity increases with floc size and low floc densities are caused by the entrapment of water in the pore spaces of flocs (Droppo et al. [75]). Accordingly, flocculation is an important mechanism for particle removal in aquatic environments such as streams and reservoirs because it alters the hydrodynamic characteristics of solids by changing their density, porosity, settling velocity and surface area (Willis and Krishnappan [73]). In the present study, flocculation altered both the effective particle size distribution and dispersion of suspended sediment along a 50 km reach of the Crowsnest River (Figure 13) and in the Oldman reservoir (Figure 14). It should be underscored that RMA4 does not presently take flocculation into account and hence the flocculation of the sediment that can occur within the reservoir was neglected. Efforts are underway to incorporate the flocculation module of RIVFLOC into RMA4.

Since elevated sediment loss is a pervasive problem associated with several types of land disturbances, the environmental significance and ecological importance of fine sediment is increasingly being recognized as a critical component of watershed management. For example, the Water Framework Directive of the European Union recognises the need for fine sediment management albeit in the context of ongoing scientific debate on the most appropriate compliance targets (European Parliament [76]; Collins and Anthony [77]). From a traditional hydraulic engineering perspective, fine sediment transport was considered to have limited importance for river morphology, channel sedimentation or reservoir management (Walling and Collins [30]). However, viewed from more transdisciplinary and especially ecological and contaminant transport perspectives, improved knowledge of the nature, mobility and transport dynamics of fine sediment and application of this information for model development and use is critical to develop and refine robust management tools for the protection of water supplies, aquatic ecology, and riparian systems under current or future climate change scenarios (Ice et al. [78]).

Improved description of the nature, mobility and transport dynamics of fine sediment is especially important in gravel bed rivers such as those in forested headwater regions with historically low sediment yields because they are often the most sensitive to minor changes in fine sediment inputs (Walling and Collins [30]; Watt et al. [79]). The significance of sediment particle size for the redistribution and fate of contaminant transport in aquatic systems has been widely reported (e.g., Ongley et al. [28]; Walling and Woodward [80]). It is a key factor controlling the form, mobility, transport, and dispersion of sediment-associated contaminants (Horowitz and Elrick [27]; Stone and Mudroch [81]; Stone and English [82]). Several studies on the effects of wildfire on nutrient export in streams show that postfire phosphorus (P) export can increase from 0.3 to more than five times greater than at pre-fire conditions (Blake et al. [83]; Silins et al. [20]). In some locations, post-fire recovery in P export can occur within two years while elsewhere a more prolonged recovery has been observed (Silins et al. [20]). For example, in an investigation of P speciation and sorption behavior of suspended sediment affected by the Lost Creek wildfire in the Crowsnest River, Emelko et al. [30] reported that sediments from the burned tributary inputs contained higher levels of bioavailable particulate P (NAIP) which were observed downstream at larger river basin scales. They also found that the potential of burned sediment to release P to the water column was significantly higher downstream of wildfire-

impacted areas compared to sediment from upstream reference (unburned) river reaches. Notably, approximately 80% of the fine post-fire sediment deposited in the downstream Oldman Reservoir originated from only ~14% of the total watershed area (Stone et al. [26]). Collectively, these previous reports emphasize the critical importance of understanding sediment transport from burned landscapes to the Crowsnest River: the elevated post-fire concentrations of bioavailable NAIP are preferentially bound to, and carried by, fine sediment, and have significant implications for the quality and treatability of water stored in the Oldman reservoir.

The mobilization and transport of sediment-associated bioavailable NAIP represents two particular water quality and treatability problems—it is critical to recognize that may manifest over different time scales. First, fine sediment entering the reservoir may remain in suspension for days; however, differences in soluble reactive phosphorus (SRP) concentrations in the river and reservoir may cause the desorption of P from the sediment into the reservoir water column, especially if the fine sediment is rich in P as has been observed after severe wildfire (Emelko et al. [21]; Watt et al. [79]). This process can occur over a period of hours (Froelich [84]). Second, over longer time periods, P-enriched sediment deposited in the reservoir may serve as a source of internal P loading to the reservoir water column because SRP can be released from anoxic sediments at the sediment-water interface (Nürnberg [85]). Accordingly, both suspended and deposited sediment can be sources of bioavailable P that can promote algal proliferation, which can lead to the associated production cyanotoxins of health concern or unpleasant taste and odour compounds (Emelko et al. [5,21]). Accordingly, the flushing frequency of the reservoir may have to be increased for risk management in response to continued loading of P-enriched fine sediment to the reservoir from wildfire-impacted upstream landscapes. Notably, this practice may likely have further implications for downstream water quality and aquatic ecology due to the more frequent release of nutrient-rich fine sediment, which may include benthic invertebrate community structure, invertebrate density, biomass, species diversity, and shifts in species composition (Silins et al. [20]; Martens et al. [86]).

## 5. Conclusions

This study demonstrates key features of a new integrated modelling framework (MOBED, RIVFLOC, RMA2, RMA4) to quantify cohesive sediment fluxes to reservoirs and inform post-fire reservoir management. While the framework was applied herein to demonstrate the prediction of fine sediment transport and fate in wildfire impacted rivers, it can be applied to inform watershed risk management and drinking water source protection by describing downstream sediment dynamics resulting from a broad range of land disturbance scenarios. The principal conclusions from this work are:

1. A new integrated modelling framework to quantify sediment fluxes to reservoirs was developed and validated. It is the first such platform for describing fine sediment transport that includes explicit description of fine sediment deposition/erosion processes as a function of bed shear stress *and* the flocculation process.
2. Bed shear stresses that prevail even at low flow conditions in the study reaches are considerably higher than the critical shear stress for deposition of fine sediment generated in the watershed. This indicates that most of the fine sediment entering the Crowsnest River from tributary inflows will be readily transported through the channel network to the downstream reservoir.
3. The process of flocculation changes the particle size distribution of suspended sediment in the water column of the Crowsnest River and influences the dispersion pattern of particles in the Oldman Reservoir because flocculation impacts the settling velocity, porosity, and density of aggregated particles (i.e., flocs). Thus, this process is an essential component of any fine (i.e., cohesive) sediment transport model, as demonstrated and validated herein. Exclusion of the flocculation process can result in underestimation of fine sediment and associated contaminant transport.

4. Deposition due to entrapment in the gravel bed study river is a possibility and this process needs to be examined further to support new process-based model parameterization.
5. Deposition patterns of sediment from wildfire-impacted landscapes were different than those from unburned landscapes because of differences in settling behaviour. These differences may lead to zones of relatively increased internal loading of phosphorus to reservoir water columns, thereby increasing the potential for algae proliferation.

**Author Contributions:** Conceptualization, B.G.K., M.S., A.L.C. and U.S.; methodology, B.G.K., M.S., U.S. and A.L.C.; software, validation, B.G.K. and C.H.S.W.; formal analysis, B.G.K. and M.S.; investigation, resources, C.H.S.W.; data curation, B.G.K., S.A.S. and U.S.; writing—original draft preparation, M.S. and B.G.K.; writing—review and editing, M.S., B.G.K., M.B.E., A.L.C. and U.S.; visualization, B.G.K.; supervision, M.S. and U.S.; project administration, M.S. and B.G.K.; funding acquisition, M.S., U.S., M.B.E. and A.L.C. All authors have read and agreed to the published version of the manuscript.

**Funding:** Field work and lab analyses were funded by NSERC Discovery Grant 481 RGPIN-2020-06963 awarded to M. Stone; Alberta Innovates Energy and Environment Solutions Grant AI-EES:2096 awarded to U. Silins, M.B. Emelko and M. Stone; and Alberta Innovates BIO Grant AI-BIO: Bio-13-009 awarded to U. Silins, M.B. Emelko and M. Stone. MBE's contribution was also enabled in part, thanks to funding from the Canada Research Chairs (CRC) Program. The contribution of ALC to this work was funded in part by the UK Biotechnology and Biological Sciences Research Council (UKRI-BBSRC) via grant BBS/E/C/000I0330. Contributions by US, CHSW, and SAS were supported by grants from ESRD/AAF (13GRFM15,15GRFFM11), AI (1865, 2343), and NSERC (216984).

**Data Availability Statement:** Data supporting reported results are with the authors can be made available upon request.

**Acknowledgments:** We gratefully acknowledge field assistance by Amanda Martens, Ashley Peter-Rennich, Samantha Karpyshin, Hans Biberhofer, Amelia Corrigan, and Kalli Herlein.

**Conflicts of Interest:** The authors declare no conflict of interest.

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
