# Peer review of "A New Framework for Modelling Fine Sediment Transport in Rivers Includes Flocculation to Inform Reservoir Management in Wildfire Impacted Watersheds"

_water, doi:10.3390/w13172319_

Round 1

Reviewer 1 Report

Dear Editor.

I have finished my review on the proposed paper “Modelling cohesive sediment transport in wildfire impacted 2 watersheds: Implications for reservoir management” water-1344293-peer-review-v1.

Summary of the manuscript:

In the proposed paper, the authors’ goal is to propose a new modelling framework to simulate flow and sediment transport. The research is a case study that was held in the Old-15 man River Watershed in Alberta, Canada, in two watersheds that was previously burned (2003) by a wildfire and a watershed unburned (as reference watershed). They investigated deposition processes in the Oldman Reservoir. The main findings of the paper was that the wildfire can alter the depositional patterns of fine sediment in the reservoir. Also there are some implications concerning the water quality and reservoir management that were influenced by the sedimentation.

General review:

The proposed paper is very well structured. It begins with an analytical Introduction with the appropriate references that helps the reader to get into the subject immediately. In Introduction there is an effort to provide previous studies with similar scientific content, which took place in the research area and in some cases in other countries. Authors describe and set very well the scientific problem and how other researchers have approached. At the end of Introduction, authors clearly state the goals of the research. The methodology is generally very interesting and well explained, so other researches could easily repeat this research methodology. The methodology is very interesting, but not novel, as it incorporates 4 already hydrogeological models. The results are very well stated and in my opinion tables and figures are easily understandable, However, I think that some changes should be made. The results scientifically explained with the use of the appropriate scientific literature. The quality of the work in Results and Discussion section is high. Conclusions are appropriate for this paper.

Points for revision:

In my opinion, the proposed paper could be characterized as a high-quality research work, complies with aims of Water. 

Nevertheless, I have some points for revision.

2.1. Study area: This section is very weak. There is no description of the three watersheds. There are no morphological and hydrographic characteristics. For example, watershed area, max and mean watersheds slope, max and mean watersheds elevation…. You can add a table. Also there is now description for the climate of the area. Mean annual precipitation, temperature, mean monthly values, max precipitation events….You can add a figure or table.

Figure 1: I propose to add a background in figure 1. It is not helpful for the reader the white background, in order to find the research area. You can add a Google image or a satellite image with good resolution.

Figure 2:  The letters of the figure are very small and the resolution low. Even if I zoom, I could not read clearly the figure.

Line 160, Table 1, line 225, table 6: Generally I am little confused about the time and duration of the research. The fire was took place at 2003. The field measurements during which years are conducted?? Table 1 inform us that field surveys were conducted at 2011 and 2015, but you did not inform us about the duration. On the other hand, in table 6 you present results for years between 2005 and 2009. It is not clear the time of surveys and the duration of these measurements. Also, it is a key point, that the research was conducted in 2011, 7 years after the wildfire. You did not say something about this, and is already known that after 3years of the fire, the forest vegetation forming soil cover, that reduce the erosion processes.

Line 232: Which is the measurement unit of the slope?? (m/m, %, degrees???)

Lines 248-253: Here you say about the Manning roughness coefficient. I propose to add three representative pictures (one for each river) that will show the rivers conditions.

Table 1: You use some devises to measure floe rate and the sediment transport. It will very interesting to show as some pictures of the devises during the field surveys.

Figure 10: Very bad quality. I cannot see nothing. The scale bar is not readable. Provide better figure.

Finally, I believe that there are some small gaps in the literature, concerning the introduction and the discussion sections.

Lines 44-51: I agree with you. However, you did not mention that there are and some other manmade disturbances that can alter (increase or decrease) the sedimentation, such new forest roads, deforestation for cropland, mineral exploitation sites, erosion control works, reforestation, abandonment of crop lands etc. (Kastridis and Kamperidou 2015, Vacca et al. 2000). I propose to add a phrase in introduction about this, and to add the proposed papers.

Lines 52-57: you should mention here that after wildfires, and during extreme rainfalls, the generation of extreme flood events is possible with parallel production and transportation of huge amounts of sediments (López-Vicente et al. 2020, Malmon et al. 2007)

Discussion section: I am fine with the discussion. However, I noticed that you did not refer anywhere in the paper the role of post fire erosion and flood control works (log check dam, log erosion barriers (LEBs), contour branch barriers, contour trenching, mulching etc.). I suppose that in your research area were not constructed these kind of works. But in many countries around the world, these works is the first priority after a wildfire.  This works are already known that significantly reduce the sediment transportation (Aristeidis and Vasiliki 2015, Wagenbrenner et al. 2006). Add a phrase about these works and add the propose literature. It is necessary to provide a solution that will reduce the production and transportation of suspended sediment.

Aristeidis, K., Vasiliki, K. Evaluation of the post-fire erosion and flood control works in the area of Cassandra (Chalkidiki, North Greece). J. For. Res. 26, 209–217 (2015). https://doi.org/10.1007/s11676-014-0005-9

  1. López-Vicente, J. González-Romero, M.E. Lucas-Borja 2020. Forest fire effects on sediment connectivity in headwater sub-catchments: Evaluation of indices performance, Science of The Total Environment, Volume 732,https://doi.org/10.1016/j.scitotenv.2020.139206.

Kastridis A., Kamperidou V. (2015): Influence of land use changes on alluviation of Volvi Lake wetland (North Greece). Soil & Water Res., 10: 121-129. https://doi.org/10.17221/174/2014-SWR

Malmon, D. V., Reneau, S. L., Katzman, D., Lavine, A., and Lyman, J. (2007), Suspended sediment transport in an ephemeral stream following wildfire, J. Geophys. Res., 112, F02006, doi:10.1029/2005JF000459.

Vacca A., Loddo S., Ollesch G., Puddu R., Serra G., Tomasi D., Aru A. (2000): Measurement of runoff and soil erosion in three areas under different land use in Sardinia (Italy). CATENA, 40, 69-92  https://doi.org/10.1016/S0341-8162(00)00088-6.

Wagenbrenner, J.W., MacDonald, L.H. and Rough, D. (2006), Effectiveness of three post-fire rehabilitation treatments in the Colorado Front Range. Hydrol. Process., 20: 2989-3006. https://doi.org/10.1002/hyp.6146

Reviewer 2 Report

The current manuscript provided a modeling framework to model sediment transport in a wildfire-impacted watershed. The writing is good, and the reviewer had the following comments:

1) This paper aims to provide a new modeling framework for sediment modeling. However, the title did not highlight such a focus. The reviewer recommends editing the title to reflect this focus of the paper.

2) The novelty of this manuscript should be the combination of the 4 models to achieve the goal of modeling fine sediment transport in rivers. However, the introduction spent little space to investigate prior work using similar approaches. The introduction mentioned “standalone” models such as SWAT and MIKE but did not talk about any example of a modeling framework. Why do we need to combine multiple models? Why do standalone models not good for such tasks? The introduction needs to be expanded in these regards.

3) Section 2.1: More information on the experimental watersheds is needed. For example, do they all have the same slope? If the slopes are different among the watersheds, it is hard to distinguish the effect of wildfire on sediment transport.

4) Resolution and text size of Fig. 2 need to be improved.

5) Section 2.2.6: Calibration of MOBED was mentioned here, without much detail. The reviewer would like to see more details in this part, such as the scope of the calibration data, whether validation was also performed, and the accuracy of calibration, etc. It is worth mentioning that figure numbers appear to be off in this section.

6) Fig. 9 and associated text: The reviewer would like to know the accuracy of the RIVFLOC calibration.

Round 2

Reviewer 1 Report

Dear Authors.

Thank you for the provided responses. 

Reviewer 2 Report

Thank you for providing a revised manuscript and response. The reviewer is satisfied and had no further questions.